# Internet-Purchased Sodium Azide Used in a Fatal Suicide Attempt: A Case Report and Review of the Literature

**DOI:** 10.3390/toxics11070608

**Published:** 2023-07-13

**Authors:** Lisa T. van der Heijden, Karen E. van den Hondel, Erik J. H. Olyslager, Lutea A. A. de Jong, Udo J. L. Reijnders, Eric J. F. Franssen

**Affiliations:** 1Department of Pharmacy & Pharmacology, Antoni van Leeuwenhoek/The Netherlands Cancer Institute, 1066 CX Amsterdam, The Netherlands; 2Division of Pharmacology, Antoni van Leeuwenhoek/The Netherlands Cancer Institute, 1066 CX Amsterdam, The Netherlands; 3Department of Forensic Medicine, GGD Amsterdam, 1066 CX Amsterdam, The Netherlands; 4Department of Clinical Pharmacy, Expert Center Gelre-iLab, Gelre Hospitals, 1066 CX Apeldoorn/Zutphen, The Netherlands; 5Department of Clinical Pharmacy, OLVG Hospital, 1066 CX Amsterdam, The Netherlands; e.j.f.franssen@olvg.nl

**Keywords:** sodium azide, postmortem toxicology, suicide

## Abstract

There has been a significant increase in sodium azide intoxications since the 1980s. Intoxications caused by sodium azide are becoming increasingly prevalent in the Netherlands as a result of its promotion for the purpose of self-euthanasia. The mechanism of toxicity is not completely understood but is dose-dependent. The presented case describes a suicide by sodium azide of a young woman (26 years old) with a history of depression and suicide attempts. The decedent was found in the presence of prescription medicine, including temazepam, domperidone in combination with omeprazole, and the chemical preservative sodium azide. Quantitative toxicology screening of whole blood revealed the presence of 70 µg/L temazepam (toxic range > 1000 µg/L) and 28 mg/L sodium azide (fatal range: 2.6–262 mg/L). Whole blood qualitative analysis revealed the presence of temazepam, temazepam-glucuronide, olanzapine, n-desmethylolanzapine, and acetaminophen. In circles promoting sodium azide, it is recommended to use sodium azide in combination with medications targeting sodium azide’s negative effects, such as analgesics, antiemetics, and anti-anxiety drugs. The medicines recovered at the body’s location, as well as the results of the toxicology screens, were consistent with the recommendations of self-euthanasia using sodium azide.

## 1. Introduction

Sodium azide (NaN_3_) is the conjugate base of hydrazoic acid (HN_3_) [1]. It is a white crystalline powder that is tasteless, odorless, and highly soluble in water [1]. Sodium azide transforms into hydrazoic acid when it comes into contact with water [1,2]. The compound is used in several fields, most notably as a propellant in vehicular airbags and safety chutes in airplanes [3]. Other applications of sodium azide are as a facilitator in synthetic reactions in chemical laboratories and as a preservative to inhibit microbial growth in biomedical laboratories [4].

In recent years, there has been a rise in the number of reports detailing (deadly) poisonings caused by sodium azide. Since the 1980s, there has been a significant increase in sodium azide intoxications in comparison to previous decades [2]. Continuing, there has been an increase in the number of suicides committed with sodium azide since the year 2000 [2]. In 2021, 37% of all case reports regarding sodium azide intoxications were suicide attempts [2]. The fatality of a sodium azide intoxication is approximately 50%, but increases to 92% if only suicide attempts are taken into account [2]. As a result of sodium azide’s promotion as a substance for use in self-euthanasia, the incidence of sodium azide poisonings increased in the Netherlands and surrounding countries [5,6]. It is likely that the convenience of purchasing the substance through online retailers has contributed to the increase in the usage of sodium azide as a (fatal) method of suicide [7,8].

The mechanism of toxicity of sodium azide is not completely elucidated. In the liver, sodium azide is metabolized to nitrogen oxide, which causes hypotension and arrhythmia [9,10]. In addition, sodium azide irreversibly blocks Cytochrome C oxidase by inhibiting oxidative phosphorylation, resulting in cell death [1]. Sodium azide also inhibits catalase, an enzyme responsible for the detoxification of hydrogen peroxide to water and oxygen [11], which can reduce ATP synthesis and cause oxidative stress [4]. The toxicity of sodium azide is dose-dependent [12]. Lower doses may result in nausea, vomiting, hypotension, tachycardia, and headaches, whereas higher doses can lead to prolonged hypotension, dysrhythmias, acidosis, seizures, and eventually death [13]. Lethal doses are reported to be above 700 mg or 10 mg/kg [14,15], while toxic doses are reported to be between 20 and 180 mg [15]. The average time between ingestion and death is 4.5 h [1].

The current case describes a suicide with sodium azide of a young woman with a history of depression and suicide attempts and illustrates the need for research of the epidemiology of sodium azide poisonings as well as ambiguous recommendations regarding toxicology screening.

## 2. Case Report

Early in the morning, a friend of the decedent (female, 26 years old), contacted the police after receiving an alarming text, which resembled a farewell message. The police arrived at the house of the decedent within an hour, finding several medications in the bedroom of the decedent. Several strips of temazepam 10 mg were discovered on the desk, although one of the strips was missing seven pills. In addition, there was half a strip of domperidone 10 mg on the desk, of which two tablets were missing, and a jar with omeprazole with the name of the decedent. On one of the desk’s shelves, a white container with a white crystalline substance was discovered. According to the label, it contained 25 g of sodium azide. External examination was performed by a forensic physician in collaboration with crime scene technicians and detectives. In the Netherlands, external examinations are performed by a forensic physician in cases of suspected unnatural causes of death or when the general practitioner is unavailable or unknown [16]. While routine post-mortem toxicological screening has not been implemented in all regions of the Netherlands, the collection of urine and femoral blood for toxicology screening has become increasingly part of the medical examination. A forensic autopsy is only performed when the case report contains questions regarding the cause of death (suspected crime). Clinical autopsies are not commonly performed due to their costly and time-consuming procedures [16,17].

The decedent had a history of depression and schizophrenia and was previously hospitalized for 10 weeks in an institution for adults with severe psychiatric disorders or addiction. Continuing, the decedent received outpatient treatment. The decedent had few social contacts, most of which she kept over the internet. The decedent also had a history of suicide attempts, and, according to her mother, the decedent had talked about the use of sodium azide in the past and communicated that she had ordered the compound online.

Toxicology screening was performed on whole blood and urine. The femoral-collected NaF-preserved blood sample was stored at −20 °C until analysis. Qualitative analysis of the whole blood sample revealed the presence of temazepam, temazepam-glucuronide, olanzapine, n-desmethylolanzapine, and paracetamol. Domperidone and omeprazole were not part of the toxicology screening. Acetone, ethanol, isopropanol, and methanol were not present in the whole blood sample. Additional quantitative analysis was performed on the whole blood sample for temazepam and sodium azide using validated LC-MS/MS and GC-MS methods, respectively. Except for the highest QC for sodium azide, which demonstrated a 16.2% bias, the methods were linear, and showed a bias of +/−15% and a precision less than 15% throughout the entire measuring range. The temazepam blood concentration was 70 µg/L (toxic range > 1000 µg/L), and the sodium azide blood concentration was 28 mg/L, which was above the upper limit of quantification (10 mg/L). A sodium azide concentration of 28 mg/L falls within the range of reported fatal concentrations [15]. Lastly, the toxicology screening of the urine sample was only positive for benzodiazepines.

## 3. Discussion

This case report presents a fatal intoxication with sodium azide after obtaining the compound through an online retailer. Sodium azide has been promoted as a drug for self-euthanasia in the Netherlands since 2017 [18]. After the first news report about drugs for self-euthanasia, a retailer reported increased sales of sodium azide [19]. However, the incidence of self-euthanasia with sodium azide, or other self-euthanasia drugs, is unknown [18]. The “right-to-die” organization supporting sodium azide for self-euthanasia recommended the administration of 1–2 g of sodium azide, along with pain killers, antiemetics, and medications used for insomnia or anxiety [18]. In the current case report, the decedent was found in the presence of the antiemetic domperidone and temazepam. These findings suggest the decedent followed the recommendations regarding self-euthanasia with sodium azide. Furthermore, the whole blood samples collected from the decedent were positive for acetaminophen. While sodium azide is recommended as a quick and humane manner of self-euthanasia, the recommended co-medication and the described symptoms discredit this claim [4,11,12,18].

The use of sodium azide is hazardous because there is no antidote or effective therapy for sodium azide poisoning available. Sodium azide demonstrates similarities to cyanide poising. However, neither treatment with sodium thiosulfate nor treatment with sodium nitrite improves clinical outcomes [12,13,20]. There is a hypothesis that methylene blue may prevent seizures caused by sodium azide by scavenging nitric oxide in the brain [21,22]. However, to date, no cases have been reported in which methylene blue has been successfully used as an antidote. Other case reports described a variety of therapeutic supportive care interventions, including sodium thiosulfate, intralipid, gastric lavage with activated charcoal, hydroxocobalamin, dopamine, dobutamine, methylprednisolone, calcium gluconate, insulin/glucose, sodium bicarbonate, and adrenaline [4,14,23,24]. Despite extensive and aggressive supportive care measures such as exchange transfusion or dialysis, in addition to the above-described therapeutic strategies, all of these individuals had to be resuscitated, and all were unsuccessful [4,14,23,24]. However, it is suggested that there might be a window of opportunity for the use of exchange transfusions or dialysis if applied before systemic effects [4] and in patients with no severe hemodynamic symptoms [14]. Due to the rapid onset of symptoms after a (high) dose of sodium azide, this treatment window may be limited.

Quantitative analysis of the deceased’s whole blood samples confirmed a sodium azide concentration of 28 mg/L, which is within the known range of fatal sodium azide concentrations (2.6–262 mg/L) [15]. In acute intoxications, the typical blood concentrations are around 50 mg/L [2]. The variability in reported sodium azide levels may be partially attributable to sodium azide’s instability in post-mortem biological material [2]. In vitro, the half-life of azide was 2.5 days at ambient temperature and 12 days at 0 °C [25]. Sodium azide was stable for 49 days in whole blood stored at −20 °C [26,27] and 49 days in plasma at −20 °C and −70 °C [24]. Therefore, it is recommended to store biological samples at −20 °C or −70 °C, protected from light. Neutralizing the pH of the matrix is another method to improve the stability of sodium azide [26]. The current reported sodium azide blood concentration of 28 mg/L could possibly be an underestimation, since the sample was thawed when it was received at the location of analysis. A second factor that could contribute to the reported range of sodium azide levels is its fast elimination [24]. Sodium azide has a reported half-life of 2.5 h in living individuals and may be detected approximately 12 h after ingestion [28]. The rapid clearance of sodium azide is consistent with findings from a number of case reports in which sodium azide was not detectable in plasma or blood but was quantifiable in stomach content [24,27]. In this case report, blood sodium azide concentrations were quantifiable, which is consistent with the time between sample collection and the estimated time of death (approximately 5 h).

The above discussion addresses several subjects that should be investigated further. First, the epidemiology of sodium azide intoxications is incompletely understood and should be investigated further. The incidence of self-euthanasia with sodium azide and information about how people obtain the compound could inform preventative strategies and legislation (e.g., steps are being taken in the Netherlands to increase the difficulty for private individuals to obtain sodium azide). Moreover, unambiguous recommendations regarding toxicology screening for suspected sodium azide poisonings are necessary. Lastly, a better understanding of the toxicological mechanism of sodium azide is essential for more and better-informed treatment options.

## 4. Conclusions

The presented case describes a suicide of a young woman with a history of depression and suicidal behavior using sodium azide. The medications found at the location of the body and the results of the toxicological tests were consistent with the recommendations of self-euthanasia with sodium azide. There are indications in the literature that the commercial availability of sodium azide is hazardous. More research is needed regarding the epidemiology of sodium azide poisonings, and ambiguous recommendations regarding toxicology screening are desired.

## Data Availability

Data available on request due to restrictions e.g., privacy or ethical.

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
