# Peer review of "Internet-Purchased Sodium Azide Used in a Fatal Suicide Attempt: A Case Report and Review of the Literature"

_toxics, 2023, doi:10.3390/toxics11070608_

Round 1

Reviewer 1 Report

The case report is prepared very well. It indcates a very important topic - availability of letal toxins via internet.

This case is a sign that the society needs education and information. The article will be published in open access journal, so it will have also an educational aspect. However, the scientific level is also very high.

I have no comments to the article. The introduction give a good background, the case is described in detail.

In my opinion, the article may be accepted in Toxics after carefull editorial corrections (eg. subscripts in chemical formulas).

Author Response

We thank the reviewer for their feedback. We altered the chemical formulas to contain subscript in the revised version of the manuscript. 

Reviewer 2 Report

Abstract

Line 16 - instead of „This current case” I suggest „Presented case”.

Line 18 - instead of „domperidon omeprazole” I suggest „domperidon in combination with omeprazole”.

Line 21 - double space as a punctuation mistake

Line 22 - instead of „acetominophen” I suggest „acetaminophen”.

Introduction and case report

Line 30 - instead of „NaN3” I suggest „NaN3” (3 in lower index) and instead of „HN3” I suggest „HN3” (3 in lower index).

Line 53 - double space as a punctuation mistake

Line 79 - double space as a punctuation mistake

Line 95 - instead of „1000 ug/L” I suggest „1000 µg/L” (the µ symbol rather than the letter u)

Bibliography

Line 179 - unnecessary enter disturbed and caused the “duplicated” numeration in the bibliography. I would like to suggest you removing the manually written numeration to provide transparency to the bibliography (up to 6th position of bibliography starts the “duplicated” numeration, 179/180 line).

Main suggestion – do not include non-English sources in bibliography

Authors include in bibliography appropriate number of bibliography position specific and adequate to short case report. Each of bibliography position appear in a correct context in the text. The bibliography is composed correctly, including 9 time-current references from the years 2020-2023.

The paper is characterized by the satisfactory academic and publishing standards being a well-written original paper type case study. In presented manuscript authors described the fatal sodium azide intoxication of young woman with the history of depression and suicide attempt from Netherland. In the “introduction” part of the presented manuscript authors performed the literature review and description of increasing rate of sodium azide caused suicide with the aid of the chemical substance bought through online retailers. In the „case report” chapter authors very accurately presented the state of the issue of sodium azide suicide intoxication and the legal implications of performing autopsies in Netherlands. The literature review in a part of „discussion” of aforementioned manuscript include multidimensional problem analysis beginning with difficult and complex issue of self-euthanasia concluding with limitation associated with laboratory preparation conditions and pre-analytical phase influencing factors in presented case.

Abstract

Line 16 - instead of „This current case” I suggest „Presented case”.

Line 18 - instead of „domperidon omeprazole” I suggest „domperidon in combination with omeprazole”.

Line 21 - double space as a punctuation mistake

Line 22 - instead of „acetominophen” I suggest „acetaminophen”.

Introduction and case report

Line 30 - instead of „NaN3” I suggest „NaN3” (3 in lower index) and instead of „HN3” I suggest „HN3” (3 in lower index).

Line 53 - double space as a punctuation mistake

Line 79 - double space as a punctuation mistake

Line 95 - instead of „1000 ug/L” I suggest „1000 µg/L” (the µ symbol rather than the letter u)

Bibliography

Line 179 - unnecessary enter disturbed and caused the “duplicated” numeration in the bibliography. I would like to suggest you removing the manually written numeration to provide transparency to the bibliography (up to 6th position of bibliography starts the “duplicated” numeration, 179/180 line).

Main suggestion – do not include non-English sources in bibliography

Authors include in bibliography appropriate number of bibliography position specific and adequate to short case report. Each of bibliography position appear in a correct context in the text. The bibliography is composed correctly, including 9 time-current references from the years 2020-2023.

The paper is characterized by the satisfactory academic and publishing standards being a well-written original paper type case study. In presented manuscript authors described the fatal sodium azide intoxication of young woman with the history of depression and suicide attempt from Netherland. In the “introduction” part of the presented manuscript authors performed the literature review and description of increasing rate of sodium azide caused suicide with the aid of the chemical substance bought through online retailers. In the „case report” chapter authors very accurately presented the state of the issue of sodium azide suicide intoxication and the legal implications of performing autopsies in Netherlands. The literature review in a part of „discussion” of aforementioned manuscript include multidimensional problem analysis beginning with difficult and complex issue of self-euthanasia concluding with limitation associated with laboratory preparation conditions and pre-analytical phase influencing factors in presented case.

Author Response

We thank the reviewer for their feedback. 

All textual suggestions have been corrected in the revised text.

We agree with the reviewer that English sources are preferable. In the current case report, Dutch sources were used. The Netherlands has an unique experience with sodium azide due to the active advertisement of the “right-to-die” organization compared to other countries. Therefore, Dutch sources describing the unique situation in the Netherlands was used when no English sources were available.

Author Response

We thank the reviewer for their feedback. 

Methods of samples collection and toxicology screening have been improved in the revised text.

Reviewer 4 Report

Dear authors

The proposed manuscript entitled “Internet-purchased sodium azide used in a fatal suicide attempt: a case report and review of literature” presents a very interesting work regarding a case report depicting the use of sodium azide for self-euthanasia. The manuscript is well written and scientifically sound. After clearing the following questions and minor English editing, I believe that the work merits publication in Toxics.

General questions:

1.      How were samples collected and preserved before analysis? Add this information to the text.

2.      You claim you used validated LC-MS/MS and GC-MS methods. Display the validated parameters, the methods and the equipment used.

3.      What makes you believe that the recommended co-medication and the described symptoms discredit the claim that sodium azide is a quick and humane manner of self-euthanasia?

Line 18 - domperidone,

Line 93 - LC-MS-MS

Line 95 - ug/L

Line 98 – missing a full stop

Line 105 – Change phase to: The "right to die" organizations supporting sodium azide for self-euthanasia, recommend the administration of 1-2 g of sodium azide, along with pain killers, antiemetics and medications used for insomnia or anxiety

Line 118 – Change phase to: There is a hypothesis that methylene blue may prevent seizures caused by sodium azide by scavenging nitric oxide in the brain [21,22]. However, to date, no cases have been reported in which methylene blue has been successfully used as an antidote.

Line 144 - may be

Author Response

We thank the reviewer for their feedback. 

  1. Information about sample collection and sample preservation have been added to the revised manuscript.
  2. The validated parameters of the LC-MS/MS and GC-MS have been included in the revised text.
  3. In our opinion a quick and humane manner of self-euthanasia would mean the person undergoing self-euthanasia would experience little to no discomfort during the procedure of self-euthanasia. However, as described in paragraph 3 of the introduction, symptoms of sodium azide poisoning like dysrhythmias, vomiting and seizures, suggest that a person might experience a lot of discomfort during the dying process. Furthermore, in our opinion the recommendation of co-medications to improve the experience of self-euthanasia with sodium azide also implies that the procedure itself is not without adverse events.

All textual suggestions have been incorporated in the revised version of the manuscript.

Reviewer 5 Report

Authors presented a case of  Intoxications caused by sodium azide, becoming increasingly prevalent in the Netherlands as a result of the promotion for the purpose of self-euthanasia. The mechanism of toxicity is not completely  understood but is dose-dependent. The case describes a suicide using sodium azide of a young woman (26-years) with a history of depression and suicide attempts.  Drugs were investigated  in the presence of prescription medicine including temazepam, domperidon omeprazole and the chemical preservative sodium azide. Quantitative toxicology screening of whole blood revealed the presence of temazepam 70 µg/L (toxic range > 1000 µg/L) and sodium azide 28 mg/L (fatal range: 20 2.6-262 mg/L). Whole blood qualitative analysis revealed the presence of temazepam, temazepam- glucuronide, olanzapine, n-desmethylolanzapine and acetominophen. In circles promoting sodium  azide it is recommended to use sodium azide in combination with medications targeting sodium azide’s negative effects such as analgesics, antiemetics and anti-anxiety drugs. The medicines recovered at the body’s location as well as the results of the toxicology screens, were consistent with the  recommendations of self-euthanasia using sodium azide. Keywords (sodium azide, postmortem toxicology, suicide) are pertinent to the case illustration meaning.

I would suggest Authors:

Internet-purchased sodium azide used in a fatal suicide attempt: a case report and review of literature. Please  Authors evaluate as preferable "the literature".

Authors observed: While routine post-mortem toxicological screening has  not been implemented in all regions of the Netherlands, collection of urine and femoral blood for toxicology screening has become increasingly part of the medical examination.  A forensic autopsy is only performed when the case report contains questions regarding  the cause of death (suspected crime). Clinical autopsies are not commonly performed due their costly and time-consuming procedures [16,17]. Please consider the analysis of suicide deaths by countries in this review:  Albano, G.D.; Malta, G.; La Spina, C.; et al., Toxicological Findings of Self-Poisoning Suicidal Deaths: A Systematic Review by Countries. Toxics 2022, 10, 654. https://doi.org/10.3390/toxics10110654.

Please consider the progressive order of listed references (7-8... ) may be is wrong inversion. 

Minor revisions of English quality are required.

Author Response

We want to thank the reviewer for their feedback. 

All suggestions have been included in the revised version of the manuscript. While we enjoyed the suggested review by Albano et al. We had difficulty placing it in our current case report since the review did not mention the Netherlands or sodium azide poisioning.